# Association of Health-Related Quality of Life with Overall Survival in Older Americans with Kidney Cancer: A Population-Based Cohort Study

**DOI:** 10.3390/healthcare9101344

**Published:** 2021-10-10

**Authors:** Naleen Raj Bhandari, Mohamed H. Kamel, Erin E. Kent, Carrie McAdam-Marx, Songthip T. Ounpraseuth, J. Mick Tilford, Nalin Payakachat

**Affiliations:** 1Division of Pharmaceutical Evaluation and Policy, University of Arkansas for Medical Sciences (UAMS), Little Rock, AR 72205, USA; bhandari.naleenraj@gmail.com; 2Department of Urology, University of Cincinnati, Cincinnati, OH 45221, USA; kamelme@ucmail.uc.edu; 3Department of Urology, Ain Shams University, Cairo 11566, Egypt; 4Department of Health Policy and Management, University of North Carolina at Chapel Hill, Chapel Hill, NC 27599, USA; erin.kent@unc.edu; 5University of North Carolina Lineberger Comprehensive Cancer Center, Chapel Hill, NC 27514, USA; 6Department of Pharmacy Practice and Science, University of Nebraska Medical Center, Omaha, NE 68198, USA; carrie.mcadammarx@unmc.edu; 7Department of Biostatistics, University of Arkansas for Medical Sciences (UAMS), Little Rock, AR 72205, USA; STOunpraseuth@uams.edu; 8Department of Health Policy and Management, University of Arkansas for Medical Sciences (UAMS), Little Rock, AR 72205, USA; TilfordMickJ@uams.edu

**Keywords:** kidney cancer, health-related quality of life, overall survival, longitudinal, SEER-MHOS

## Abstract

Background: Our purpose was to evaluate associations between health-related quality of life (HRQoL) and overall survival (OS) in a population-based sample of kidney cancer (KC) patients in the US. Methods: We analyzed a longitudinal cohort (*n* = 188) using the Surveillance, Epidemiology, and End Results (SEER) database linked with the Medicare Health Outcomes Survey (MHOS; 1998–2014). We included KC patients aged ≥65 years, with a completed MHOS during baseline (pre-diagnosis) and another during follow-up (post-diagnosis). We reported HRQoL as physical component summary (PCS) and mental component summary (MCS) scores and OS as number of months from diagnosis to death/end-of-follow-up. Findings were reported as adjusted hazard ratios (aHRs (95% CI)) from Cox Proportional Hazard models. Results: The aHRs associated with a 3-point lower average (baseline and follow-up) or a 3-point within-patient decline (change) in HRQoL with OS were: (a) baseline: PCS (1.08 (1.01–1.16)) and MCS (1.09 (1.01–1.18)); (b) follow-up: PCS (1.21 (1.12–1.31)) and MCS (1.11 (1.04–1.19)); and (c) change: PCS (1.10 (1.02–1.18)) and MCS (1.02 (0.95–1.10)). Conclusions: Reduced HRQoL was associated with worse OS and this association was strongest for post-diagnosis PCS, followed by change in PCS and pre-diagnosis PCS. Findings highlight the prognostic value of HRQoL on OS, emphasize the importance of monitoring PCS in evaluating KC prognosis, and contribute additional evidence to support the implementation of patient-reported outcomes in clinical settings.

## 1. Introduction

Health-related quality of life (HRQoL), a multidimensional concept representing individuals’ subjective assessment of overall wellbeing [1] is becoming a commonly measured patient-reported outcome (PRO) in oncology clinical trials [2,3,4,5]. Additionally, there is a growing interest in implementing PRO measures in routine clinical practices [3,6]. Research in the United States (US), suggests that older kidney cancer patients experience poorer physical HRQoL versus those without cancer, while there is no difference in their mental HRQoL [7,8,9]. Kidney cancer is commonly diagnosed in older individuals (≥60 years) [10], who also frequently present with several age-related comorbidities. This may result in decreased functional ability, which not only affects HRQoL but also complicates healthcare management [11].

In cancer patients, overall survival (OS) is shown to be associated with age at diagnosis, cancer type and stage, treatments, comorbidities, disability, and sociodemographic factors [12,13]. In patients with kidney cancer, survival is improving [14,15,16], and researchers have projected that kidney cancer will be managed as a chronic disease [17]. Thus, clinicians need more accurate ways to monitor disease prognosis and to manage patients’ needs, with the ultimate goal of improving long-term survival. Therefore, research in oncology has begun to evaluate the value of PROs, including HRQoL (a multidimensional concept representing an individual’s subjective assessment of overall wellbeing), in predicting clinical outcomes.

A review of the CheckMate-025 trial (nivolumab versus everolimus) revealed a positive correlation between baseline HRQoL and OS in previously treated patients with advanced kidney cancer who received nivolumab [18]. Similarly, patients with fewer symptoms [19,20,21] or better HRQoL [22,23] experienced longer progression-free survival and OS. HRQoL has also been found to predict OS in patients with lung cancer, performing better than a physician-rated performance status [24], and indicating the value of considering patients’ perspective in understanding cancer prognosis. Observational research in patients with other cancer types (including lung and ovarian cancers) [13,25,26,27], and in a general sample of older adults [28], has also demonstrated a positive association between HRQoL and long-term survival. 

Understanding the prognostic ability of HRQoL in predicting OS in kidney cancer has been limited, given the heterogeneity across studies related to the timing of HRQoL assessments. This study examined the association of HRQoL with overall survival in older kidney cancer patients aged ≥65 in the US. We hypothesized that: (a) a lower average HRQoL at baseline (pre-diagnosis) and follow-up (post-diagnosis) would be associated with an increase in hazard of death; and (b) a greater reduction in HRQoL from baseline to follow-up would be associated with an increase in hazard of death.

## 2. Methods

### 2.1. Data

This was a longitudinal, retrospective study (Appendix A) using the 1998–2014 Surveillance, Epidemiology, and End Results (SEER) linked with Medicare Health Outcomes Survey (MHOS) data. The SEER is a US-based cancer registry that currently covers ~34% of the population [29]. SEER-MHOS contain information about incident cancer diagnosis, demographics, and date of death from the SEER and sociodemographic factors, health problems, and PROs, including HRQoL from the MHOS. The MHOS is a survey of 1000–1200 Medicare Advantage (MA) beneficiaries, who are randomly selected each year from participating MA plans. They are invited to participate in a baseline MHOS and the respondents are resurveyed after two years with follow-up MHOS. Detailed information about SEER-MHOS is available elsewhere [30].

### 2.2. Study Population

We included patients with a primary diagnosis of kidney cancer (International Classification of Diseases for Oncology, Third Edition Code = C64.9) who had at least one survey prior to (referred to as baseline or pre-diagnosis in this study, T0) and at least one survey after diagnosis of kidney cancer (referred to as follow-up or post-diagnosis in this study, T1). Among patients with >1 survey, the most recent pre-diagnosis survey was selected as their baseline while the latest survey post-diagnosis was included as their follow-up survey. Patients with a missing date of diagnosis of kidney cancer or those aged <65 years during the month of diagnosis were excluded.

### 2.3. Study Measures

*OS,* the outcome, was measured as the number of months from the diagnosis of kidney cancer until death (all cause) or the end of the follow-up. Patients alive at the end of their follow-up were censored.

*HRQoL* was measured using Short Form-36 (SF36, 1998–2005) or Veterans RAND 12-item (VR12, 2006 onwards) health surveys, depending on the MHOS cohort. HRQoL was measured at pre-diagnosis, post-diagnosis, and as a change from pre-diagnosis to post-diagnosis of kidney cancer. These surveys are comparable and measure the same eight scales of HRQoL (physical functioning, role-physical, bodily pain, general health, mental health, role-emotional, social functioning, and vitality) [31]. These instruments provide two summary scores: physical component summary (PCS) and mental component summary (MCS), representing physical and mental health, respectively. The values for PCS, MCS, and eight scales range from 0 to 100 where a higher score indicates better HRQoL. For these scores, a 3-point difference represents a minimal clinically important difference (MCID), which is defined as the smallest difference/change in a measure that would be considered a clinically meaningful difference to patients [32,33]. Another measure of HRQoL used in this study was health utility (HU) that represents the “value assigned to different health states,” [1], estimated using published algorithms [34]. The HU scores range from 0.29 to 1, where a score of 1 indicates “full health”, and a difference of 0.03 points represents the MCID [35].

Several cancer sites, including kidney cancer, have been shown to be associated with reductions in HRQoL scales [8,9], which are associated with OS in lung cancer patients [25,26]. Therefore, in this study, PCS/MCS were the two main exposures of interest while secondary exposures of interest included the eight scales and HU scores.

*Covariates* The covariates considered for multivariable analyses in this study included sociodemographic characteristics, comorbidities, kidney cancer-related factors, duration between the pre-diagnosis survey and kidney cancer diagnosis (T0 models), and/or duration between diagnosis and the post-diagnosis survey (T1 and change models). Covariates that were common across T0, T1, and change (T1-T0) multivariable models were gender, race, age at diagnosis, stage at diagnosis, tumor grade, type of initial treatment, and geographic location during kidney cancer diagnosis. Following covariates could vary with time, so they were measured at pre-diagnosis (for T0 models) and at post-diagnosis (for T1 and change models) separately: education status, marital status, annual household income, smoking status, and the number of self-reported comorbidities. The number of months between the pre-diagnosis survey and kidney cancer diagnosis (“Months_T0_Dx”) was included as an additional covariate in T0 models. Similarly, the number of months between kidney cancer diagnosis and the post-diagnosis survey (“Months_Dx_T1”) was included as an additional covariate in T1 models. Lastly, both “Months_T0_Dx” and “Months_Dx_T1” were included as additional covariates in the change models.

### 2.4. Statistical Analyses

We used Chi-square or Student’s *t* tests to compare patient-level characteristics between patients who died during the study period vs. those who did not. The Kaplan–Meier analysis was used to determine the median OS. Using separate multivariable Cox Proportional Hazards models, adjusted associations between HRQoL (PCS, MCS, eight scales, and HU at T0, T1, and as a change from T0 to T1) and OS were determined, and the covariate-adjusted hazard ratios (aHRs) and 95% confidence intervals were reported. 

We also evaluated whether including HRQoL measure in these models provides a prediction advantage for OS using likelihood ratio tests [36] for nested models and computed generalized R-squared (R^2^) [37]. A statistically significant likelihood ratio test indicates that a model with HRQoL better explains the outcome. Similarly, the higher the generalized R^2^ (range: 0–1), the stronger the association. We conducted these tests for PCS, MCS, and HU models.

#### Sensitivity Analyses

Because OS could be confounded by patients’ pre-diagnosis HRQoL, in *sensitivity analysis I (SA-I)*, we included pre-diagnosis HRQoL as an additional covariate in all T1 and change models. In *sensitivity analysis II (SA-II)*, we determined if study findings were sensitive to selection bias due to the selection of a healthier cohort in the main analyses as patients were required to have both pre-diagnosis and post-diagnosis surveys. To do this, we repeated all T0 models from the main analyses on all older patients with kidney cancer who had one pre-diagnosis survey regardless of whether or not they had a post-diagnosis survey. In *sensitivity analysis III (SA-III)*, we also repeated all T1 models from the main analyses on all older KC patients who had one post-diagnosis survey regardless of whether or not they had a pre-diagnosis survey.

## 3. Results

One hundred and eighty-eight (*n* = 188) patients with kidney cancer with a median (interquartile range (IQR)) follow-up of 79 (54–120) months were included and analyzed in this study (Figure 1). The average age at diagnosis of kidney cancer was 74.9 ± 5.5 years. We noted 88 (47%) deaths with a median (IQR) OS of 107 (45–165) months (Table 1).

### 3.1. Patient Characteristics

Patients who survived versus those who died had statistically significant differences in baseline characteristics (Table 1). Surviving patients were 2 years younger at diagnosis versus those who died (74.0 ± 4.8 vs. 75.9 ± 6.0, *p* = 0.017). A greater proportion of surviving patients were nonsmokers (84.8% vs. 64.1%, *p* = 0.004). None of the surviving patients were diagnosed with distant kidney cancer disease versus 14.6% of those who died (*p* < 0.001).

Except in MCS, no differences in average baseline HRQoL measures between the two groups were observed (Appendix A). Patients who survived reported better average MCS versus those who died (54.8 ± 9.0 vs. 50.8 ± 11.4, *p* = 0.009). Average post-diagnosis HRQoL in patients who survived was significantly better than those who died and the difference in all measures exceeded MCIDs. The average within-person change in HRQoL from pre-diagnosis to post-diagnosis between the two groups was significantly different for PCS, HU, role-physical, and general health. Patients who died had greater reductions in HRQoL (>MCIDs) from pre-diagnosis to post-diagnosis versus those who survived.

### 3.2. Association between HRQoL and OS

During baseline, every measure of HRQoL, except the scales bodily pain, mental health, and vitality, was significantly associated with OS. The aHRs ranged from 1.08 to 1.09, indicating that a 0.03-point lower average HU or 3-point lower average PCS/MCS during pre-diagnosis was associated with an 8–9% increase in hazard of death.

During follow-up, every measure of HRQoL, except role-emotional scale, was significantly associated with OS. The aHRs ranged from 1.11 to 1.21, which represents an 11–21% increase in hazard of death.

Among all the measures indicating the change in patients’ HRQoL from pre-diagnosis to post-diagnosis, only changes in PCS and scales for physical functioning, social functioning, and vitality were significantly associated with OS. Their aHRs ranged from 1.07 to 1.10, representing a 7–10% increase in hazard of death (Figure 2 and Appendix A).

We also noted that addition of HRQoL information provided a predictive advantage for OS. Of the likelihood ratio tests comparing models that included HRQoL versus those that did not include HRQoL for HU, PCS, and MCS during pre-diagnosis, post-diagnosis, and change from pre-diagnosis to post-diagnosis, all except one were statistically significant (Table 2). Likewise, we observed an increase in generalized R^2^ in 8/9 tests (Table 2). Both approaches indicate that models including HRQoL better capture the relationship between OS and predictors. 

### 3.3. Sensitivity Analyses

In *SA-I* adjusting for pre-diagnosis HRQoL, the magnitudes of aHRs for the change models were significantly different (Figure 3 and Appendix A, *SA-I*), from those observed in the main analyses. This indicates the importance of including baseline HRQoL in an explanatory model when determining the relationship between change in HRQoL and OS. However, the magnitudes of aHRs observed for follow-up models in *SA-I* were identical to those observed for follow-up models in the main analyses.

When exploring the risk of selection bias by removing requirements to have follow-up MHOS (*SA-II*, *n =* 1055), some level of selection bias was evident given that most point estimates for HRQoL measures were no longer significantly associated with OS. However, PCS and social functioning remained significantly associated with OS but with a smaller magnitude (aHR = 1.04–1.03). When the requirement to have baseline MHOS was removed (*SA-III*, *n =* 966), the findings were consistent with those observed in the main analyses, although the magnitudes of the associations were smaller (aHR = 1.05–1.10) (Appendix A).

## 4. Discussion

To our knowledge, this is the first study to quantify the association between HRQoL measurements recorded during pre-diagnosis and post-diagnosis of kidney cancer and OS in older patients with kidney cancer using a population-based database, where the findings suggest a significant association between HRQoL and OS. The strengths of these associations were generally strong; however, they differed with respect to the timing and domain of HRQoL assessment. Patients’ HRQoL during post-diagnosis was more strongly associated with OS than HRQoL from pre-diagnosis or HRQoL changes (T1-T0), which was also largely supported in sensitivity analysis III but with weaker associations. This pattern was mostly consistent across both primary (PCS and MCS) and secondary (eight scales and HU) exposures of interest. These findings would be useful to clinicians/geriatricians in better understanding the prognosis of kidney cancer, and they will aid in providing appropriate healthcare to improve patients’ long-term survival.

Most findings noted in this study are consistent with previous clinical trials [18,22,23,24] and similar studies in patients with other cancers [25,26]. We observed a significant increase (8–9%) in hazard of death with clinically meaningful lower average PCS/MCS during baseline, which was also observed in sensitivity analysis II for PCS but with a lower magnitude. However, in patients with lung [26] or advanced ovarian [27] cancers, pre-diagnosis HRQoL was not associated with OS. In this study, a 3-point lower average PCS/MCS during post-diagnosis was associated with an 11–21% (or 7–10% in sensitivity analysis III) increase in hazard of death, which was greater than that observed in lung cancer patients [26]. Additionally, in this study, when longitudinal changes within the same patients were analyzed, a 3-point reduction in PCS from pre-diagnosis to post-diagnosis was significantly associated with OS (10% increased risk). These findings potentially indicate that in patients with kidney cancer, impairment in physical health has a stronger association with OS vs. mental health. Assessing patients’ long-term physical HRQoL could help in recognizing those with poorer physical health and may provide opportunities for early clinical intervention. Preliminary evidence in other cancers suggests benefits of non-pharmacological interventions (e.g., physical activity) in the improvement of HRQoL and fatigue [38,39,40]; however, such evidence in patients with kidney cancer is rare [41,42].

In this study, 7/8 HRQoL scales measured post-diagnosis were significantly associated with OS, where reduced HRQoL was associated with 6–17% increased hazard of death, which was also observed in sensitivity analysis III but with smaller magnitude. However, upon evaluating patients’ change in HRQoL from pre- to post-diagnosis, clinically meaningful reductions were only identified in physical functioning, social functioning, and vitality scales, which were associated with a significantly higher (7–10%, or 10–17% in sensitivity analysis I) hazard of death. This is of clinical relevance because the average reductions (from pre- to post-diagnosis) in these scales were ≥MCIDs among patients who died. By definition, a meaningful reduction in (a) physical functioning indicates limitations in performing physical activities such as bathing and dressing; (b) social functioning is indicative of physical and emotional problems interfering with patients’ normal social activities; and (c) vitality suggests feelings of tiredness and being worn out [43]. Longitudinal monitoring of these domains could help in identifying patients at greater risk for death. Enhancing the delivery of and referral to clinical resources to improve physical functioning, reduce fatigue, and bolster social support may potentially result in improved survival among these patients.

The findings from likelihood ratio tests and generalized R^2^ in this study provide evidence in favor of including a measure of HRQoL when estimating kidney cancer patients’ likelihood of survival, emphasizing the prognostic value of HRQoL on OS. We also observed that only the association between changes in HRQoL (pre- to post-diagnosis) and OS was sensitive to baseline HRQoL, while the association between follow-up HRQoL and OS was not. This may have happened considering that each patient may start at a different HRQoL during baseline, which has a greater association on the change measure than the follow-up measure of HRQoL. The relationship of kidney cancer diagnosis with patients’ health may also be different given their baseline status of health that is observed in the change measure of HRQoL, indicating the importance of having multiple HRQoL assessments when evaluating long-term impacts of HRQoL changes on OS.

There are limitations to this study worth noting. The generalizability of findings may be subject to selection bias due to a healthier study cohort. However, our sensitivity analyses in a larger cohort of patients with kidney cancer confirmed the association between HRQoL and OS, potentially with weaker associations. Additionally, we compared baseline sociodemographic factors, PCS and MCS for KC patients who had both baseline and follow-up surveys versus those who only had a baseline survey and did not find any significant differences (data not shown). The study findings may not be generalizable to patients with other cancers, younger age, or those KC patients enrolled in fee-for-service Medicare. There is inconclusive evidence about the similarity of characteristics between patients enrolled in fee-for-service vs. Medicare Advantage health plans [7,44,45,46,47]. Moreover, residual confounding may be present in our study as we were unable to control for use of chemotherapy, severity of comorbidities, etc., due to the unavailability of information.

## 5. Conclusions

The lower average HRQoL pre-diagnosis and post-diagnosis of kidney cancer and a greater decline in PCS from pre- to post-diagnosis of KC were associated with increased hazard of death. Post-diagnosis HRQoL was a stronger predictor of overall survival compared to HRQoL measured prior to diagnosis or as a change from pre- to post-diagnosis, which was also confirmed in sensitivity analyses in this study. Findings highlight the prognostic value of HRQoL on OS and emphasize the importance of monitoring physical health in assessing the prognosis of kidney cancer in older Americans. Given the increased emphasis placed on patient-reported outcomes, HRQoL monitoring should be added to kidney cancer patients’ long-term care plans and can be used in developing prognostic criteria for these patients [13,48]. 

## Figures and Tables

**Figure 1 healthcare-09-01344-f001:**
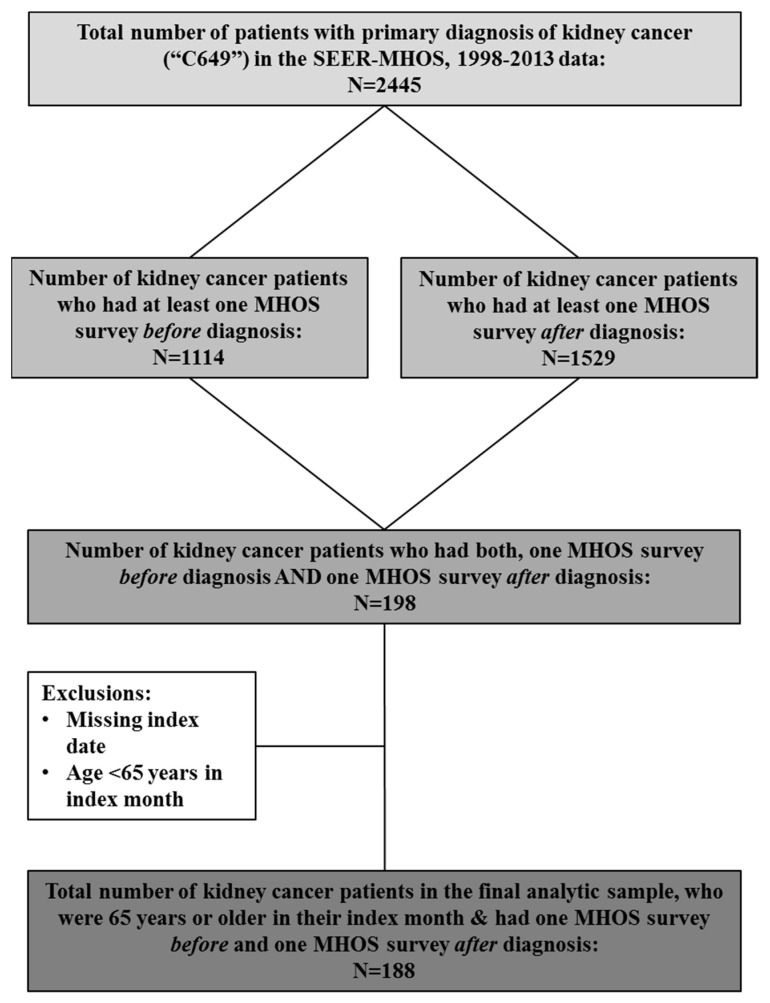
Study sample selection.

**Figure 2 healthcare-09-01344-f002:**
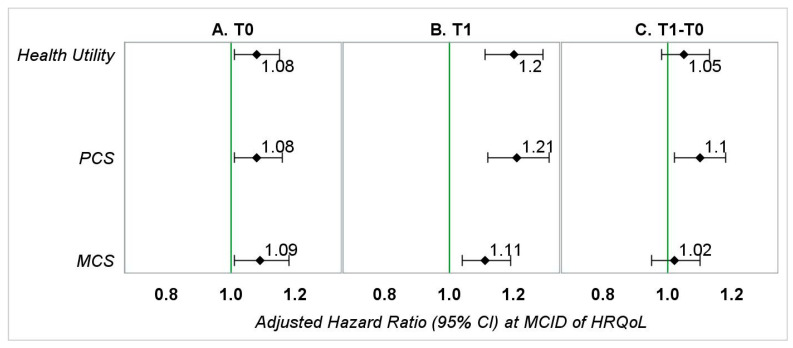
Adjusted hazard ratios (95% CI) representing the association between HRQoL (at MCID) and OS in patients with kidney cancer—main analyses. PCS—physical component summary; MCS—mental component summary. (**A**) Hazard ratios represent the association between *baseline HRQoL* (0.03-point or 3-point reduction) and overall survival adjusted for gender, race, education (T0), marital status (T0), annual household income (T0), smoking status (T0), geographic region, number of comorbid conditions (T0), age at diagnosis, stage of kidney cancer at diagnosis, tumor grade, treatment type, and months between the T0 survey and kidney cancer diagnosis. (**B**) Hazard ratios represent the association between follow-up *HRQoL* (0.03-point or 3-point reduction) and overall survival adjusted for gender, race, education (T1), marital status (T1), annual household income (T1), smoking status (T1), geographic region, number of comorbid conditions (T1), age at diagnosis, stage of kidney cancer at diagnosis, tumor grade, treatment type, and months between diagnosis of kidney cancer and the T1 survey. (**C**) Hazard ratios represent the association between change in HRQoL (0.03-point or 3-point reduction) from baseline to follow-up and overall survival adjusted for gender, race, education (T1), marital status (T1), annual household income (T1), smoking status (T1), geographic region, number of comorbid conditions (T1), age at diagnosis, stage of kidney cancer at diagnosis, tumor grade, treatment type, months between the T0 survey and kidney cancer diagnosis, and months between diagnosis of kidney cancer and the T1 survey.

**Figure 3 healthcare-09-01344-f003:**
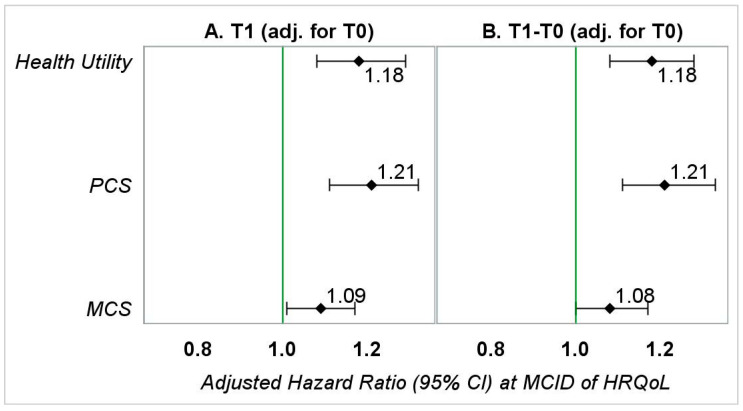
Adjusted hazard ratios (95% CI) representing the association between HRQoL (at MCID) and OS in patients with kidney cancer—sensitivity analyses I. PCS—physical component summary; MCS—mental component summary. (**A**) Hazard ratios represent the association between *follow-up HRQoL* (0.03-point or 3-point reduction) and overall survival adjusted for gender, race, education (T1), marital status (T1), annual household income (T1), smoking status (T1), geographic region, number of comorbid conditions (T1), age at diagnosis, stage of kidney cancer at diagnosis, tumor grade, treatment type, months between diagnosis of kidney cancer and the T1 survey, and baseline HRQoL. (**B**) Hazard ratios represent the association between change in *HRQoL* (0.03-point or 3-point reduction) from baseline to follow-up and overall survival adjusted for gender, race, education (T1), marital status (T1), annual household income (T1), smoking status (T1), geographic region, number of comorbid conditions (T1), age at diagnosis, stage of kidney cancer at diagnosis, tumor grade, treatment type, months between the T0 survey and kidney cancer diagnosis, months between diagnosis of kidney cancer and the T1 survey, and baseline HRQoL.

**Table 1 healthcare-09-01344-t001:** Characteristics of patients with kidney cancer.

Variables	Total ^†^ (*n* = 188), *n* (col %)	Group, *n* (col %) ^†^		*p*-Value
		Survived (*n* = 99, 52.7%)	Died (*n* = 89, 47.3%)	
**Age at Diagnosis (Dx, years)**				
Mean ± SD	74.9 ± 5.5	74.0 ± 4.8	75.9 ± 6.0	0.017
Median (IQR)	74.0 (71.0–79.0)	73.0 (71.0–78.0)	75.0 (72.0–80.0)	0.030 ^‡^
**Male**	101 (53.7)	51 (51.5)	50 (56.2)	0.522
**White**	136 (72.3)	67 (67.7)	69 (77.5)	0.132
**Education**				0.582
High school or lower	120 (63.8)	61 (61.6)	59 (66.3)	
Some college or higher	65 (34.6)	37 (37.4)	28 (31.5)	
**Marital Status**				0.433
Married	120 (63.8)	66 (66.7)	54 (60.7)	
Non-married	64 (34.1)	32 (32.3)	32 (35.9)	
**Annual Household Income**				0.164
<$30,000	80 (42.6)	36 (36.4)	44 (49.4)	
≥$30,000	54 (28.7)	30 (30.3)	24 (27.0)	
Do not know or missing	54 (28.7)	33 (33.3)	21 (23.6)	
**Current Smoker**				0.004
Yes	14 (7.5)	<11 (<11.1)	<11 (<11.1)	
No	141 (75.0)	84 (84.8)	57 (64.1)	
Do not know or missing	33 (17.5)	<11 (<11.1)	>12 (>13.5)	
**Geographic Region**				0.014
West/Midwest	117 (62.2)	58 (58.6)	59 (66.3)	
South	44 (23.4)	30 (30.3)	14 (15.7)	
Northeast	27 (14.4)	11 (11.1)	16 (18.0)	
**Stage at Diagnosis**				<0.001
Localized	135 (71.8)	79 (79.8)	56 (62.9)	
Regional	36 (19.2)	19 (19.2)	17 (19.1)	
Distant	13 (6.9)	-	13 (14.6)	
**Tumor Grade**				0.018
I/II	100 (53.2)	62 (62.6)	38 (42.7)	
III/IV	39 (20.7)	19 (19.2)	20 (22.5)	
Unknown	49 (26.1)	18 (18.2)	31 (34.8)	
**Treatment Type**				0.159
Nephron sparing	63 (33.5)	32 (32.3)	31 (34.8)	
Radical nephrectomy	122 (64.9)	67 (67.7)	55 (61.8)	
**Number of Comorbidities**				
Mean ± SD	2.9 ± 1.9	3.0 ± 1.9	2.9 ± 1.8	0.518
Median (IQR)	3.0 (2.0–4.0)	3.0 (2.0–4.0)	2.0 (1.0–4.0)	0.583 ^‡^
**Months between T0 Survey and Dx**				
Mean ± SD	17.4 ± 18.0	17.6 ± 19.2	17.1 ± 16.7	0.850
Median (IQR)	13.9 (6.6–20.9)	11.7 (5.6–20.4)	15.1 (6.9–21.0)	0.421 ^‡^
**Months between Dx and T1 Survey**				
Mean ± SD	23.7 ± 26.6	29.1 ± 27.3	17.6 ± 24.7	0.003
Median (IQR)	15.4 (8.0–24.4)	18.4 (11.0–38.7)	10.9 (5.3–21.0)	<0.001 ^‡^

Note: A few cell sizes have been suppressed to protect patient identity; **^†^** proportions may not always add up to 100% due to missing values; ^‡^ based on Kruskal–Wallis test. Number of comorbidities was a summary variable to indicate total number of self-reported diagnoses of the following conditions: hypertension, angina pectoris or coronary artery disease, congestive heart failure, myocardial infarction, other heart conditions, stroke, emphysema or chronic obstructive pulmonary disease or asthma, ulcerative colitis or Crohn’s Disease or inflammatory bowel disease, arthritis of the hip or knee, arthritis of the hand or wrist, sciatica, and diabetes.

**Table 2 healthcare-09-01344-t002:** Likelihood ratio tests and generalized R^2^ scores indicating the importance of including HRQoL in assessing OS.

Model	Likelihood Ratio Test		Generalized R^2^	
	Likelihood Ratio Statistic	*p*-Value	Full Model	Reduced Model
Baseline-Health Utility	28.2	<0.001	0.32	0.29
Baseline-PCS	5.2	0.023	0.31	0.29
Baseline-MCS	5.1	0.024	0.31	0.29
Follow-up-Health Utility	43.8	<0.001	0.46	0.38
Follow-up-PCS	36.2	<0.001	0.46	0.38
Follow-up-MCS	8.2	0.004	0.41	0.38
Change-Health Utility	44.5	<0.001	0.40	0.38
Change-PCS	16.3	<0.001	0.40	0.38
Change-MCS	0.4	0.539	0.38	0.38

Abbreviations: PCS—physical component summary; MCS—mental component summary. The variables in *full baseline* models included baseline HRQoL, age at diagnosis, number of comorbidities, months between baseline survey and diagnosis, gender, race, education, marital status, annual household income, smoking status, geographic region, stage at diagnosis, tumor grade, and treatment type. The variables in *reduced baseline* models included all from the respective full model, except baseline HRQoL. The variables in *full follow-up* models included follow-up HRQoL, age at diagnosis, number of comorbidities, months between diagnosis and follow-up survey, gender, race, education, marital status, annual household income, smoking status, geographic region, stage at diagnosis, tumor grade, and treatment type. The variables in *reduced follow-up* models included all from the respective full model, except follow-up HRQoL. The variables in *full change* models included change in HRQoL, age at diagnosis, number of comorbidities, number of months between baseline survey and diagnosis, months between diagnosis and follow-up survey, gender, race, education, marital status, annual household income, smoking status, geographic region, stage at diagnosis, tumor grade, and treatment type. The variables in *reduced change* models included all from the respective full model, except change in HRQoL.

## Data Availability

The database used in this study cannot be made publicly available as it was obtained from the United States National Cancer Institute (https://healthcaredelivery.cancer.gov/seer-mhos/) under an approved data use agreement.

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
