# Peer review of "Association of Health-Related Quality of Life with Overall Survival in Older Americans with Kidney Cancer: A Population-Based Cohort Study"

_healthcare, 2021, doi:10.3390/healthcare9101344_

Round 1

Reviewer 1 Report

This study investigates the relationship between HRQoL and overall survival in patients with kidney cancer. The study is based on registry data and concludes that a lower HRQoL is associated with reduced survival. The authors have conducted extensive sensitivity analyses to explore the potential biases in their analyses. Overall, the paper is very well written and uses adequate statistical methodology. I only have a couple of comments:

Comments:

  1. The biggest shortcoming in my view is the restriction that two measurements of HRQoL must be available for patients to be included in the analysis. This means that patients must survive long enough for the second measurement to be collected. While the authors mention this issue in the manuscript and provide sensitivity analyses to investigate its impact, I believe it should be stated even more clearly in the discussion/conclusion section. For example, a statement such as “Post-diagnosis HRQoL was a stronger predictor of overall survival” should not be made without cautioning.

  2. Just making sure that I understood correctly: The models adjusted for HRQoL contained each of the scores (e.g. PCS, MCS) separately and these were coded such that a one-unit change represents this 3-point difference. Is that correct? Perhaps this could be stated more clearly in the methods section to avoid confusion.

  3. How did you deal with missing values in the Cox models?

  4. Some care should be taken with respect to causal conclusions. As stated by the authors, they cannot rule out residual bias due to unmeasured confounding and the analyses are not adjusted for baseline differences between the groups. Yet the statements on page 6/7 imply a causal effect of HRQoL on OS, stating that changing the former might improve survival. I think this statement should be re-phrased more carefully, given that the results are based on observational data and the analyses have not been adjusted for that (except through conditioning on the available covariates).

  5. Related to my first point, the final paragraph of the discussion states with respect to selection bias: “However, our sensitivity analyses in a larger cohort of patients with kidney cancer confirmed the association between HRQoL and OS.” However, the associations in the adjusted model (SA II) are much weaker and not many of them are significant any more. Thus, I recommend to phrase this more carefully.

Reviewer 2 Report

This paper evaluated associations between health-related quality of life (HRQoL) and overall survival (OS) in population-based sample of kidney cancer (KC) patients in the US and they concluded that reduced HRQoL was associated with worse OS and this association was strongest for post-diagnosis PCS, followed by change in PCS and pre-diagnosis PCS. I do have some comments as listed below in the order noted.

Comment 1:

The quality of the data set is very important, especially in a SEER-MHOS population-based people. For this reason, please clarify the inclusion criteria (such as: ICD-9 CM diagnosis code) and exclusion criteria of Study Sample Selection in the Methods section.

Comment2:

Please clarify and define MCID in the study.

Comment 3:

Please provide the mean ± standard deviation and median [IQR] for the Health Utility, PCS, and MCS at T0 and T1 time points.

Reviewer 3 Report

The authors' efforts are appreciated. The study is well prepared and follows an appropriate methodology. However, there are some questions that need to be answered.

- The hypothesis is actually two hypotheses when the title speaks of only one.
- It is not clear how you have calculated the sample size and from where you have drawn the final sample.
- The word "covarietis" I think is not in the correct position.
- In all cancers early diagnosis is an important factor in survival. The age of the patient is an important factor as research has shown. Kidney cancer has a 5-year survival rate of 75%. They have studied the association of cancer with other disease and the influence on quality of life. Many of these patients have several diseases at the same time and are even immunosuppressed so it may be normal for their quality of life to decrease.
- The most important association with cancer survival in their study was being a smoker or non-smoker. The authors believe that these risk factors are more important than taking antidepressant medication in improving quality of life.
- A pilot study was conducted before reaching the conclusion they have reached. They do not believe that it is normal for quality of life to decrease with an illness is something obvious or that they really bring with the conclusion that is totally novel to other previous studies.
